# Therapeutic Resistance Models and Treatment Sequencing in Advanced Prostate Cancer

**DOI:** 10.3390/cancers15215273

**Published:** 2023-11-03

**Authors:** Zachary A. Schaaf, Shu Ning, Amy R. Leslie, Masuda Sharifi, Xianrui Han, Cameron Armstrong, Wei Lou, Alan P. Lombard, Chengfei Liu, Allen C. Gao

**Affiliations:** 1Department of Urologic Surgery, University of California Davis, Sacramento, CA 95817, USA; zaschaaf@ucdavis.edu (Z.A.S.); xshning@ucdavis.edu (S.N.); arleslie@ucdavis.edu (A.R.L.); msharifi@ucdavis.edu (M.S.); xrahan@ucdavis.edu (X.H.); cmarms@ucdavis.edu (C.A.); weilou@ucdavis.edu (W.L.); aplombard@ucdavis.edu (A.P.L.); cffliu@ucdavis.edu (C.L.); 2UC Davis Comprehensive Cancer Center, University of California Davis, Sacramento, CA 95817, USA; 3Department of Biochemistry and Molecular Medicine, University of California Davis, Sacramento, CA 95616, USA; 4VA Northern California Health Care System, Sacramento, CA 95655, USA

**Keywords:** prostate, cancer, resistance, enzalutamide, abiraterone, darolutamide, apalutamide, docetaxel, olaparib

## Abstract

**Simple Summary:**

Castration-resistant prostate cancer (CRPC) treatments include next-generation anti-androgen therapies (NGATs), taxane therapy, and PARP inhibitors (PARPi). However, resistance often occurs across and within therapeutic classes, which can complicate sequential treatment options. We developed acquired resistant models to study therapeutic resistance. Our findings indicate that while NGAT-resistant cells are cross-resistant to other NGATs, they remain sensitive to taxanes and olaparib. Cells resistant to docetaxel display cross-resistance to cabazitaxel and olaparib but respond to NGATs. Olaparib-resistant cells are cross-resistant to other PARPi but still sensitive to NGATs and docetaxel. Our research underscores the significance of rationale drug sequencing in CRPC treatment and underlying mechanisms of resistance.

**Abstract:**

Current common treatments for castration-resistant prostate cancer (CRPC) typically belong to one of three major categories: next-generation anti-androgen therapies (NGAT) including enzalutamide, abiraterone acetate, apalutamide, and darolutamide; taxane therapy represented by docetaxel; and PARP inhibitors (PARPi) like olaparib. Although these treatments have shown efficacy and have improved outcomes for many patients, some do not survive due to the emergence of therapeutic resistance. The clinical landscape is further complicated by limited knowledge about how the sequence of treatments impacts the development of therapeutic cross-resistance in CRPC. We have developed multiple CRPC models of acquired therapeutic resistance cell sublines from C4-2B cells. These include C4-2B MDVR, C4-2B AbiR, C4-2B ApaR, C4-2B DaroR, TaxR, and 2B-olapR, which are resistant to enzalutamide, abiraterone, apalutamide, darolutamide, docetaxel, and olaparib, respectively. These models are instrumental for analyzing gene expression and assessing responses to various treatments. Our findings reveal distinct cross-resistance characteristics among NGAT-resistant cell sublines. Specifically, resistance to enzalutamide induces resistance to abiraterone and vice versa, while maintaining sensitivity to taxanes and olaparib. Conversely, cells with acquired resistance to docetaxel exhibit cross-resistance to both cabazitaxel and olaparib but retain sensitivity to NGATs like enzalutamide and abiraterone. OlapR cells, significantly resistant to olaparib compared to parental cells, are still responsive to NGATs and docetaxel. Moreover, OlapR models display cross-resistance to other clinically relevant PARP inhibitors, including rucaparib, niraparib, and talazoparib. RNA-sequencing analyses have revealed a complex network of altered gene expressions that influence signaling pathways, energy metabolism, and apoptotic signaling, pivotal to cancer’s evolution and progression. The data indicate that resistance mechanisms are distinct among different drug classes. Notably, NGAT-resistant sublines exhibited a significant downregulation of androgen-regulated genes, contrasting to the stable expression noted in olaparib and docetaxel-resistant sublines. These results may have clinical implications by showing that treatments of one class can be sequenced with those from another class, but caution should be taken when sequencing drugs of the same class.

## 1. Introduction

Prostate cancer (PCa) is one of the most common types of cancer among men, with an estimated 1.4 million cases diagnosed annually worldwide [1]. Initial therapies utilized for treatment that work to reduce systemic androgens such as surgical castration/resection or anti-androgen therapeutics prove to be effective, however, in as little as just 5 years almost 20% of patients experience Castration-Resistant Prostate Cancer (CRPC) [2,3,4]. Treatment of CRPC constitutes additional androgen signaling targeted therapy and widely used chemotherapeutics [5]. Commonly used drugs in the class of androgen signaling targeted therapy including enzalutamide (Enza), abiraterone (Abi), apalutamide (Apa), and darolutamide (Daro), in chemotherapy include docetaxel (Tax) and cabazitaxel, and molecular-targeted therapy include PARP inhibitors such as olaparib (OlapR). These treatments have shown improvements in patient outcomes [6].

Androgen signaling targeted therapy for CRPC has shown efficacy due to the importance of the AR axis in the occurrence and development of PCa. Abiraterone acts as an AR signaling dampener by effectively inhibiting cytochrome P450 c17 (CYP17), which is responsible for intracrine androgen biosynthesis and has been effective in increasing patient overall survival in the CRPC setting [7]. Working more directly on the AR axis, Enza is a second-generation androgen signaling inhibitor with AR inhibition mechanisms that include blocking of androgen-AR binding, AR nuclear translocation, and DNA binding [8], with well-established efficacy in patients with CRPC [9]. In addition, other recently developed AR inhibitors including Apa [10], as well as Daro, exhibit a higher AR binding affinity than either Enza [11].

Treatment for CRPC with chemotherapy is primarily taxane-associated [6]. Docetaxel is an anti-mitotic taxane-based chemotherapeutic that promotes microtubule stabilization, induces apoptosis, and is commonly used for PCa patients that progress to CRPC [12,13]. Molecular-targeting based therapy in the CRPC setting by PARP inhibitors includes olaparib, a small potent molecule inhibitor of PARP1 and PARP2 enzymes [14], which causes an accumulation of DNA damage in rapidly dividing cancer cells. As of 2020, olaparib was approved for use in DNA repair-deficient patients with advanced CRPC [15,16].

Despite advancements in the therapeutic landscape for CRPC, therapeutic resistance often occurs [17]. Various mechanisms of acquired therapeutic resistance arise [6,12,18,19], which are multifaceted and can vary across cancer types [18,20]. Resistance to mentioned drugs can be established via cellular changes including drug concentration, cell death evasion, DNA repair, inflammatory signaling changes, and other ways [6]. Previously, point mutations, as well as other aberrations in AR or the AR signaling pathway reprogramming have been observed in enzalutamide and other AR signaling inhibitor resistance [6,21]. Autophagy has also been implicated in enzalutamide resistance in CRPC [22]. Abiraterone resistance has been shown through increased expression of CYP17A1 and other key players in androgen production [23,24]. Common clinical forms of olaparib resistance include mutations re-enabling DNA (Homologous Recombination) repair [25,26]. Olaparib resistance has been described by our group in CRPC cells, with an ability to override the G2/M cell cycle checkpoint and tolerate higher levels of olaparib-induced DNA damage [27]. Docetaxel resistance has been described in PCa cancer through the utilization of muti-drug resistance proteins (MDRP) such as drug efflux pumps [6,12,28]. Recently, our group reported activation of the ABCB1 drug transporter amplicon as a mediator of taxane and specifically docetaxel resistance in CRPC [29].

Cross-resistance across various therapeutics is an important issue as the emergence of resistance increases the likelihood of a need for sequential treatment with alternative therapeutics as certain drugs lose efficacy. The use of the anti-androgen signaling drugs Enza or Abi has been observed to show reduced efficacy when used in sequence regardless of order [30,31]. Our group has previously reported that ABCB1 gene upregulation mediated cross-resistance between taxanes and olaparib [32]. As for NGAT resistance, we have reported that both Enza and Abi-resistant cells display cross-resistance with the other NGAT therapy but not with Tax [33].

Unfortunately, there still exists a gap in knowledge regarding a more comprehensive view of how derived resistant CRPC cells differ from their parents, as well as how they respond to various therapeutics. We have developed a series of resistant cell sublines to androgen signaling targeted therapy from parental C4-2B cells including C4-2B MDVR, C4-2B AbiR, C4-2B ApaR, and C4-2B DaroR resistance to enzalutamide, abiraterone, apalutamide, and darolutamide. In addition, we also generated TaxR and OlapR C4-2B cells resistant to docetaxel and olaparib (Figure 1A). In this study, we performed RNA-sequencing analysis to define pathways common or unique to the resistant cells and to understand the sequencing treatment of these drugs for prostate cancer. Our goals are to identify potential therapeutic targets and develop strategies to overcome drug resistance, as well as the rationale and evidence behind their sequencing in the treatment of prostate cancer.

## 2. Methods

### 2.1. Cell Lines and Culture 

NGAT-resistant (NGAT-R) lines were generated from parental C4-2B cells as described previously by our group with resistance to enzalutamide (MDVR) [34], abiraterone acetate (AbiR) [34], and apalutamide (ApaR) [35]. Likewise, docetaxel-resistant (TaxR) [36] cells and olaparib-resistant OlapR [27] cells were also generated from the C4-2B background through exposure to docetaxel and olaparib, respectively, as previously described by our group. Resistance to darolutamide cell subline was generated from C4-2B cells chronically exposed to increasing concentrations of darolutamide (5 μM–40 μM) over passages in media over 12 months to create the darolutamide-resistant C4-2B cell subline (DaroR).

C4-2B cells were maintained in RPMI 1640 media supplemented with 10% fetal bovine serum, 100 IU penicillin, and 0.1 mg/mL streptomycin (supplemented media). Resistant cell lines were maintained with supplemented media with the drug they exhibit resistance to. MDVR cells were correspondingly maintained with 20 μM enzalutamide, AbiR cells with 10 μM abiraterone acetate, ApaR cells with 20 μM apalutamide, DaroR cells with 20 μM darolutamide, OlapR cells with 5 μM olaparib, and TaxR cells with 5 nM docetaxel. All cells were maintained at 37 °C in a humidified incubator with 5% carbon dioxide and routinely tested for mycoplasma with a PCR detection kit (Abcam, Cambridge, UK).

### 2.2. Growth Assays

To assess the sensitivity of C4-2B parental and derived resistant lines to the therapeutics described, all cell groups were utilized in growth assays with varying doses of the mentioned xenobiotics. A total of 2 × 10^4^ cells were plated in 24-well cell culture plates (Corning, Corning, NY, USA) and treated with varying doses of either NGATs, olaparib or docetaxel for 48 h before counting to analyze comparative growth. Cell numbers were counted with Z1 particle counter (Beckman Coulter, Brea, CA, USA).

### 2.3. RNA Sequencing

Parental C4-2B as well as resistant derivative lines MDVR, AbiR, ApaR, DaroR, OlapR, and TaxR were harvested from culture. TriZOL reagent (Invitrogen, Waltham, MA, USA) was used on harvested cells to extract total RNA. Quantity and purity were assessed using RNA Nano 6000 Assay Kit, on the (Agilent, Santa Clara, CA, USA) Bioanalyzer 2100 system. Sequencing libraries were then generated from RNA using the NEB Next Ultra RNA Library Prep Kit (New England BioLabs, Ipswich, MA, USA) and index codes were added. The cDNA fragments 150–200 bp in length were then selected using the AMPure XP system (Beckman Coulter, Brea, CA, USA). Following this, the index-coded samples were further clustered using PE Cluster Kit cBot-HS (Illumina, San Diego, CA, USA) and sequenced on the Illumina platform. Paired-end clean reads were aligned to reference genome assembly (GRCh38/hg38) using the Spliced Transcripts Alignment to a Reference software (STAR version 2.7.3), and number of fragments per kilobase of transcript per million fragments mapped (FPKM) was calculated for genes and subsequently used for enrichment analysis.

### 2.4. Gene Set Enrichment Analysis (GSEA)

RNA sequencing results enable the comparison of differential expression across multiple genes. Transcriptome data were then used to determine overarching trends in sequencing results with Gene Set Enrichment Analysis (GSEA version 4.1.0). GSEA software (Broad Institute, Cambridge, MA, USA) provided by the MSigDB (http://software.broadinstitute.org/gsea/index.jsp (accessed on 1 Feburary 2023)) was utilized in conjunction with the Molecular Signature Database (MSigDB) collection of gene sets. Enrichment analysis was conducted with curated sets provided by MSigDB, as well as Gene Ontology (GO), the Kyoto Encyclopedia of Genes and Genomes (KEGG), Pathway Interaction Database (PID), WikiPathways (WP), and Reactome Pathways. Heatmaps were generated based on normalized enrichment score (NES) when comparing resistant cell lines with parental C4-2B. NES lower than one across all resistant lines were excluded from the data presentation. NES > 1 with *p* < 0.05 is considered as significant enrichment of pathway activity in resistant cell lines.

### 2.5. Statistical Analysis

Cell growth assays were performed in triplicates and from three independent experiments. GraphPad Prism 8.0 was used for data normalization, analysis, and visualization. Cell number was normalized as a percent of control (equivalent concentration of DMSO) to compare growth across groups. Therapeutic concentration was then transformed to a log scale, and a nonlinear regression trendline was derived under proper parameters (log(inhibitor) vs. normalized response—variable slope) in GraphPad. IC_50_ was derived and reported as well as the 95% confidence interval (95%CI).

## 3. Results

### 3.1. Characterization of Treatment-Induced Resistant Sublines

We have developed a series of resistant cell sublines from parental C4-2B cells, which demonstrate resistance to androgen signaling targeted therapies (Figure 1A). These sublines include MDVR [34], AbiR [34], ApaR [35], and DaroR, which are resistant to enzalutamide, abiraterone, apalutamide, and darolutamide, respectively. Additionally, we derived TaxR [36] and OlapR [27] sublines from C4-2B cells that are resistant to docetaxel and olaparib, respectively.

To thoroughly characterize these cell sublines, we determined their respective IC_50_ values for the drugs to which they are resistant. The NGAT-resistant (NGAT-R) cell lines, for instance, exhibit robust resistance compared to C4-2B parental cells. The IC_50_ values, compared between parental and resistant sublines, are as follows: MDVR to enzalutamide (C4-2B vs. MDVR, 18.84 μM vs. 41.64 μM) (Figure 1B), AbiR to abiraterone acetate (C4-2B vs. AbiR, 10.35 μM vs. 15.04 μM) (Figure 1C), ApaR to apalutamide (C4-2B vs. ApaR, 24.7 μM vs. 47.96 μM) (Figure 1D), and DaroR to darolutamide (C4-2B vs. DaroR, 11.08 μM vs. 19.07 μM) (Figure 1E). In addition, the OlapR subline shows extensive resistance to olaparib (C4-2B vs. OlapR, 1.823 μM vs. 57.87 μM) (Figure 1F), and the TaxR subline exhibits immense resistance to docetaxel (C4-2B vs. TaxR, 1.042 nM vs. 106.9 nM) (Figure 1G).

### 3.2. Intra-Class Cross-Resistance among NGAT-Rs

Derivative lines demonstrate resistance to the growth inhibition exerted by various therapeutics with which they are cultured. However, it remained uncertain whether individual NGAT-R lines would also exhibit cross-resistance to other anti-androgen signaling therapeutics within the same category, even without prior exposure to these therapeutics. Our findings confirm that NGAT-R lines do indeed exhibit varying levels of resistance to other NGATs.

The difference in growth with enzalutamide between parental C4-2B and NGAT-R cells was evident. The enzalutamide IC_50_ values are as follows: 18.84 μM for C4-2B, 41.64 μM for MDVR, 35.41 μM for AbiR, 24.26 μM for ApaR, and 27.81 μM for DaroR (Figure 2A). NGAT-resistant lines also demonstrated resistance to abiraterone acetate when compared to the parental line, with IC_50_ values of 10.35 μM for C4-2B, 13.71 μM for MDVR, 15.04 μM for AbiR, 12.82 μM for ApaR, and 16.61 μM for DaroR cells (Figure 2C).

This trend in cell response was also evident with apalutamide and darolutamide. The apalutamide IC_50_ values were 24.7 μM for C4-2B, 50.82 μM for MDVR, 43.97 μM for AbiR, 47.96 μM for ApaR, and 45.7 μM for DaroR (Figure 2B). The darolutamide IC_50_ values were 11.08 μM for C4-2B, 17.1 μM for MDVR, 13.24 μM for AbiR, 14.96 μM for ApaR, and 19.07 μM for DaroR (Figure 2D). Consequently, we observed that NGAT-R cells exhibit intra-class cross-resistance to anti-androgen therapies.

In our previous studies, we demonstrated cross-resistance between docetaxel-resistant TaxR cells and cabazitaxel, as well as between cabazitaxel-resistant CabR cells and docetaxel [29]. Similarly, olaparib-resistant OlapR cells exhibited cross-resistance to other PARP inhibitors such as rucaparib, niraparib, and talazoparib [27]. In conclusion, our data collectively suggest that cross-resistance is a prevalent phenomenon within each class of drugs.

### 3.3. Cellular Response across Therapeutic Classes

In addition to NGATs, other common therapeutics for advanced prostate cancer include chemotherapy taxanes, such as docetaxel, and targeted therapy PARP inhibitors like olaparib. Having established that cross-resistance occurs within each class of these drugs, we proceeded to investigate the effects of one class of drug on cells resistant to another class. We utilized resistant sublines MDVR, TaxR, and OlapR, with each representing resistance to a different class of drug.

Our data revealed that the IC_50_ for enzalutamide on resistant MDVR cells is 41.64 μM. In contrast, the IC_50_ for parental C4-2B, TaxR, and OlapR cells were 18.84 μM, 20.20 μM, and 23.21 μM, respectively (Figure 3A). This indicates that both TaxR and OlapR cells retain sensitivity to enzalutamide, similar to their parental C4-2B cells.

We then subjected the parental C4-2B cells and their resistant sublines MDVR, TaxR, and OlapR to varying doses of docetaxel to evaluate growth inhibition. Notably, all resistant sublines, except TaxR, exhibited sensitivity to docetaxel, with IC_50_ values of 1.042 nM for C4-2 B, 0.699 nM for MDVR, and 1.367 nM for OlapR, while TaxR had an IC_50_ of 106.9 nM (Figure 3C). This indicates a retained sensitivity of MDVR and OlapR cells to docetaxel.

Furthermore, olaparib inhibited growth at lower concentrations in naïve C4-2B and derivative MDVR lines compared to other derivative cells. The olaparib IC_50_ values were 1.82 μM for C4-2B, 3.31 μM for MDVR, 57.87 μM for OlapR, and 25.56 μM for TaxR (Figure 3B). These findings suggest that CRPC cells with prior exposure to docetaxel exhibit a reduced response to olaparib compared to naïve cells or those exposed to enzalutamide.

In conclusion, our data underscore the existence of varied responses among different classes of drugs. This suggests that sequencing treatments with different classes of drugs could benefit patients who relapse after treatment with one class of these drugs.

### 3.4. Molecular Changes in Resistant Cell Sublines

We aimed to unravel the potential molecular alterations linked with each resistant cell subline. Therefore, we employed gene set enrichment analysis on the transcriptomes of resistant cell sublines, comparing them with their parental C4-2B cells. We carried out the GSEA analysis with curated sets provided by the Gene Ontology Consortium (GO: www.geneontology.org (accessed on 1 February 2023)) (Figure 4). Our findings highlighted a notable downregulation in the expression of genes associated with AR signaling across all derivative lines when compared to the shared parental C4-2B. We subsequently explored energy metabolism. The analysis revealed an upregulation in genes associated with oxidative phosphorylation-based ATP synthesis across all resistant lines, except for TaxR, where a marked downregulation was observed. Next, glycolysis-associated genes showed an upregulation in OlapR and TaxR cells, with a moderate upregulation also detectable in ApaR and DaroR cells. Conversely, MDVR and AbiR cells were characterized by a downregulation in the expression of glycolysis-related genes.

An increase in gene expression of cholesterol metabolism was discernible across NGAT-R lines and docetaxel-resistant TaxR yet absent in OlapR cells when benchmarked against C4-2B. DNA repair-associated gene expression exhibited an elevation in OlapR and, marginally, in TaxR cells. Evaluating cell signaling pathways, we noticed an upregulation of NOTCH signaling in MDVR, AbiR, ApaR, and OlapR, albeit absent in DaroR or TaxR cells. A consistent downregulation of cell cycle checkpoint-signaling proteins characterized all NGAT-R cell lines. In contrast, OlapR and intriguingly, TaxR, exhibited augmented checkpoint signaling. When assessing genes modulating the inflammation response, we observed an amplified expression in MDVR, AbiR, ApaR, and OlapR cells. Uniquely, DaroR emerged as the sole NGAT-R displaying a reduced inflammatory response, a trait shared, to some extent, by TaxR. In the context of apoptotic signaling, increased expression of regulatory genes was identifiable in MDVR, AbiR, ApaR, and TaxR. In contrast, DaroR and OlapR showed diminished expression in the corresponding gene set. Focusing on drug metabolism-based genes, increased protein expression was evident in MDVR, AbiR, ApaR, and TaxR, whereas DaroR and OlapR exhibited a downregulation.

Expression of genes involved in inflammatory signaling was notably increased in TaxR and across NGAT-R lines except for DaroR which exhibited downregulation along with OlapR. Gene expression relating to the NOTCH signaling pathway was also upregulated across NGAT-R sublines aside from DaroR, OlapR was upregulated, and TaxR was downregulated. Similarly for STAT signaling pathway expression, NGAT-R sublines were upregulated aside from DaroR, which was downregulated as well as OlapR. TaxR showed increased expression related to STAT signaling. WNT signaling-related expression was analyzed, with the highest upregulated exhibited in MDVR cells followed by other NGAT-R sublines except for DaroR which was again downregulated. OlapR WNT signaling expression was decreased compared to parental, whereas TaxR expression was increased comparatively. Lastly, the GO gene set for TGFb was utilized on transcriptome data and unveiled a slight upregulation in MDVR and TaxR cells, and a decrease in expression for other sublines including AbiR, ApaR, DaroR, and OlapR.

#### 3.4.1. Androgen Receptor Signaling Expression Is Downregulated in NGAT-R but Not in TaxR and OlapR Cells

We delved deeper into the expression of genes associated with androgen signaling and uncovered distinct patterns. A trend of downregulation in androgen signaling-related gene expression was apparent in cell lines resistant to NGATs. Conversely, cell lines resistant to either olaparib or docetaxel did not display a similar degree of downregulation in androgen-related genes (Figure 5A).

#### 3.4.2. DNA Repair Gene Expression

DNA repair is needed for genomic maintenance, and associated pathways are commonly hindered through mutation or transcription aberration in the cancer phenotype. This offers genomic instability to support rapid proliferation but increases susceptibility to DNA-damaging agents or events [37]. In our examination of NGAT-resistant cell lines, we found a consistent downregulation in the expression of DNA repair-associated genes. In contrast, the olaparib-resistant cells showed a slight upregulation in the expression of genes associated with DNA repair (Figure 5A).

#### 3.4.3. Apoptotic Signaling

In an intriguing yet paradoxical phenomenon, it is recognized that cancers of differing malignancy levels exhibit variable apoptotic signaling. Notably, high-grade cancers associated with a poorer prognosis often exhibit elevated levels of apoptosis [38]. In our analysis of derivative resistant lines, MDVR cells stood out, displaying a heightened expression of pro-apoptotic genes. This pattern of upregulation was not consistently observed in other cell lines. Notably, DaroR cells exhibited a general downregulation across the explored gene sets, underscoring the complex and diverse nature of apoptotic signaling in cancer cells (Figure 5A).

#### 3.4.4. Cell Cycle Gene Expression

Cell cycle-related gene expression and the various cellular components that they encode are often altered in the cancer landscape [39]. Our study reveals a notable trend; all derivative lines that have been subjected to chronic therapeutic exposure, except for OlapR cells, exhibit a decreased expression of genes associated with cell proliferation and the cell cycle (Figure 5A).

#### 3.4.5. Alterations of Oxidative Phosphorylation/Glycolysis Pathway Expression

Changes in cancer cell metabolism offer views into potential mechanisms through which cells fuel viability and growth [40]. To further understand this in CRPC, we aimed to identify the differential expression of genes that are integral to specific metabolic processes. Notably, cell lines resistant to NGATs and olaparib showed an increased expression of genes associated with oxidative phosphorylation compared to their parental line. In contrast, cell lines resistant to docetaxel exhibited a decreased transcription of genes involved in promoting cellular oxidative phosphorylation (Figure 5B).

We also examined glycolysis, another pivotal pathway in cellular metabolism. Our findings revealed an upregulation in this pathway in both olaparib and docetaxel-resistant lines. However, for those resistant to NGATs there was a slight downregulation, with DaroR exhibiting the most pronounced decrease (Figure 5B).

#### 3.4.6. Upregulation of Fatty Acid/Cholesterol Gene Expression in NGAT-R Cells

Other major metabolic pathways that affect prostate cancer progression include lipid metabolism [41]. In our study of NGAT-resistant lines, we observed an upregulation of genes associated with fatty acid and cholesterol metabolism. Notably, the AbiR subline displayed the most prominent increase in the expression of genes related to cholesterol metabolism (Figure 5B).

#### 3.4.7. Xenobiotic Metabolism/ABC Transporter Expression in Resistant Cell Lines

A common mechanism of therapeutic resistance is achieved through drug inactivation or clearance [42]. As previously reported by our group as well as others, ABC transporter expression can infer therapeutic resistance in cancer cells [29,32,36]. Overall, xenobiotic clearance mechanisms were upregulated in AbiR and TaxR cells. As previously reported by our group as well as others, ABC transporter expression can infer therapeutic resistance in cancer cells [29,32,36]. In our ongoing studies, we observed that xenobiotic clearance mechanisms were notably upregulated in AbiR and TaxR cells. A closer examination revealed a varied but consistent upregulation of ABC transporter expression across all resistant cell lines. Intriguingly, TaxR cells displayed the most significant increase in ABC transporter expression, aligning with our expectations and underscoring the transporter’s role in conferring drug resistance (Figure 5B).

#### 3.4.8. Inflammatory Signaling in Resistant Cells

Inflammatory signaling is commonly observed in cancer cells and can help fuel malignant progression as well as therapeutic resistance [43,44,45]. Although the cell lines in our study were examined in culture and not within a complex multicellular tumor microenvironment, notable changes in gene expression associated with inflammatory signaling were still evident. All NGAT-R cell lines, particularly MDVR, exhibited upregulated expression compared to the parental C4-2B cells, indicating enhanced inflammatory signaling. An exception to this pattern was observed in DaroR cells, which displayed an aberration in inflammation-mediated pro-carcinogenic pathways.

The resistant cell lines OlapR and TaxR presented variable expression patterns when compared with the parental line, underscoring the complexity and diversity of inflammatory responses among different cell lines (Figure 5C).

#### 3.4.9. Analysis of Cell Signaling Pathways in Cancer Progression

Signaling pathways, including mTOR [46], KRAS [47], STAT [48], and other integrin/receptor-mediated pathways like NOTCH, WNT, TGF-beta, and hedgehog signaling, play crucial roles in cancer progression [43]. In our study, we observed that the overall expression related to cell signaling in derivative lines was generally upregulated compared to the parental C4-2B cells. Specifically, the MDVR cell line showed a consistent and broad upregulation across the various signaling pathways explored. In contrast, DaroR cells displayed a substantial decline in the expression of a majority of cell signaling gene sets (Figure 5D).

## 4. Discussion

The goals of this study were to identify potential therapeutic targets and develop strategies to overcome drug resistance, as well as the rationale and evidence behind their sequencing in the treatment of prostate cancer. We have established multiple CRPC models of acquired therapeutic resistance using cell sublines derived from C4-2B cells (Figure 1). These include C4-2B MDVR, C4-2B AbiR, C4-2B ApaR, C4-2B DaroR, C4-2B TaxR, and C4-2B OlapR, which are resistant to enzalutamide, abiraterone, apalutamide, darolutamide, docetaxel, and olaparib, respectively [27,34,35,36]. These models are implemented for analyzing gene expression and assessing responses to various treatments.

The systematic exploration of IC_50_ values across diverse therapeutics and cell lines elucidates a complex yet patterned landscape of resistance, one that underscores the evolution of cancer cells’ adaptability and resilience against therapeutic interventions. The intra-class cross-resistance to growth inhibition exhibited by resistant sublines within one class of drugs, as demonstrated by elevated enzalutamide, abiraterone acetate, apalutamide, and darolutamide IC_50_ values, reflects a common adaptive mechanism. This complicates therapeutic strategies but also illuminates potential pathways for innovative interventions. This ingrained cross-resistance, corroborated by our prior findings on TaxR and CabR cells’ mutual resistance [32] and OlapR’s resistance to various PARP inhibitors [27], posits a pivotal challenge to therapeutic strategies.

The retained sensitivity of NGAT-resistant lines to taxane and PARP inhibitors, and vice versa, suggests a delineation in the adaptive mechanisms invoked by different therapeutic classes. This specificity in adaptation, echoed in the responses to docetaxel across the resistant sublines and the varied sensitivity to olaparib, illuminates a landscape for therapeutic exploitation. These insights suggest the possibility of a strategic alternation and combination of therapeutic classes to overcome the cells’ adaptive mechanisms and pave the way for tailored therapeutic sequencing, potentially enhancing efficacy while mitigating resistance.

Our in-depth study into cancer cell sublines resistant to NGATs, olaparib, and docetaxel has revealed significant insights into the adaptive mechanisms underlying therapeutic resistance. We have uncovered a complex interplay of altered gene expressions affecting signaling pathways, energy metabolism, and apoptotic signaling, which are central to the evolution and progression of cancer. One notable finding is the divergent modulation of androgen signaling among the resistant cell sublines. NGAT-resistant sublines showed significant downregulation of androgen-related genes, contrasting with the stable expression observed in olaparib and docetaxel-resistant sublines.

In the context of energy metabolism, we identified upregulation of oxidative phosphorylation genes in NGATs and olaparib-resistant sublines, contrasting with their downregulation in docetaxel-resistant subline cells. This metabolic adaptability of cancer cells points towards potential targets for therapeutic interventions aiming to disrupt energy production pathways and impair cell viability. Our findings also emphasized the complex and variable nature of cell signaling dynamics under therapeutic pressure, marked by distinct adaptive responses among the resistant cell sublines. Moreover, we found an increased pro-apoptotic gene expression in MDVR cells, a characteristic feature of high-grade, aggressive cancers. ABC transporters emerged as significant contributors to therapeutic resistance to docetaxel. Their upregulation, particularly in TaxR cells, underscores an urgent need to develop inhibitory strategies to enhance the efficacy of current treatments.

In conclusion, our findings delineate a dynamic and complex landscape of cancer cell adaptability, with each therapeutic modality eliciting specific and varied adaptive responses. These insights are instrumental for not only understanding the multifaceted nature of therapeutic resistance but also for developing innovative, tailored strategies for sequencing treatments to counter these adaptations effectively (Figure 6). While more research is needed to determine the optimal sequencing strategy for these drugs in treatment-resistant prostate cancer, these cell lines can help guide the development of new treatments and improve outcomes for patients with advanced disease.

## 5. Conclusions

These experimental results argue not only for therapeutic resistance, but also the existence of intra-cross-resistance and inter-cross-response. Our findings indicate that while NGAT-resistant cells are cross-resistant to other NGATs, they remain sensitive to taxanes and olaparib. Cells resistant to docetaxel display cross-resistance to cabazitaxel and olaparib but respond to NGATs. Olaparib-resistant cells are cross-resistant to other PARPi but still sensitive to NGATs and docetaxel. Transcriptomic analyses revealed alterations in gene expression across treatment naïve as well as resistant cell types. These data underscore the significance of rationale drug sequencing in CRPC treatment and underlying mechanisms of resistance.

## Figures and Tables

**Figure 1 cancers-15-05273-f001:**
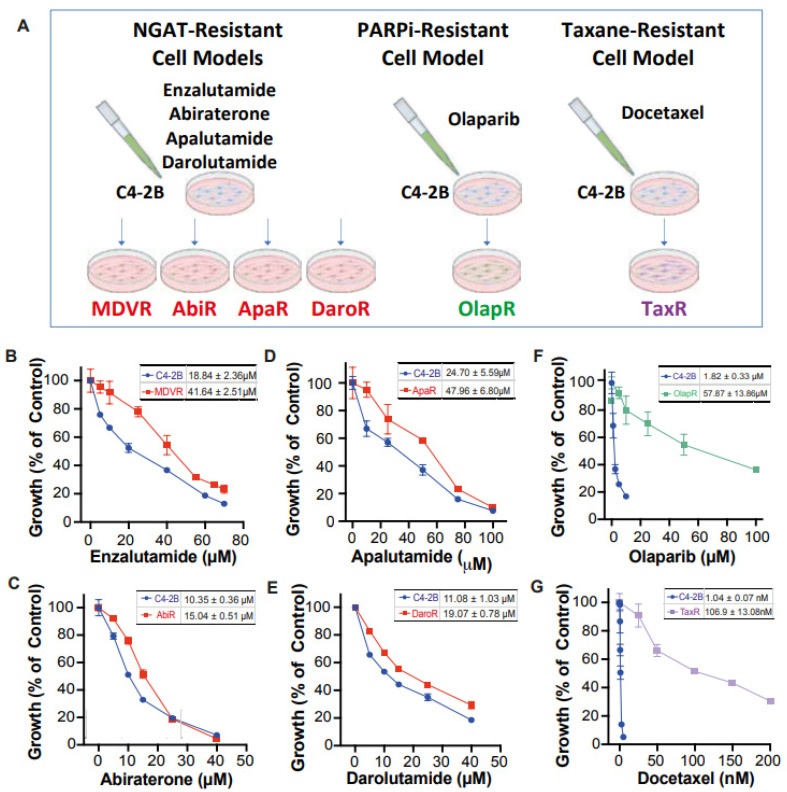
Creation of C4-2B-derived therapeutic-resistant CRPC cell lines. (**A**) Schematic illustration depicting C4-2B-derived therapeutic-resistant line series creation through chronic exposure. Dose-response curve under varying concentrations of therapeutic for both C4-2B parental and respective derived lines (**B**) MDVR (**C**) AbiR (**D**) ApaR (**E**) DaroR (**F**) OlapR (**G**) TaxR. IC_50_ with 95% CI was presented correspondingly for each cell line.

**Figure 2 cancers-15-05273-f002:**
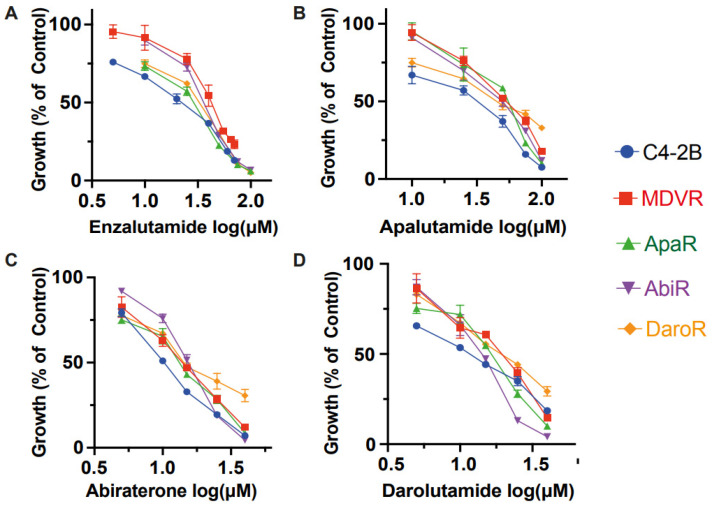
Intra-class resistance across NGAT-Resistant lines. NGAT-resistant cells display resistance to growth inhibition by other therapies within the same class. C4-2B and NGAT-R cells (MDVR, ApaR, AbiR, and DaroR) were plated and treated with corresponding doses of (**A**) enzalutamide (**B**) apalutamide (**C**) abiraterone (**D**) darolutamide for 48 h, and cell numbers were counted and quantified as percentage changes to control groups and presented as the mean (±S.D.).

**Figure 3 cancers-15-05273-f003:**
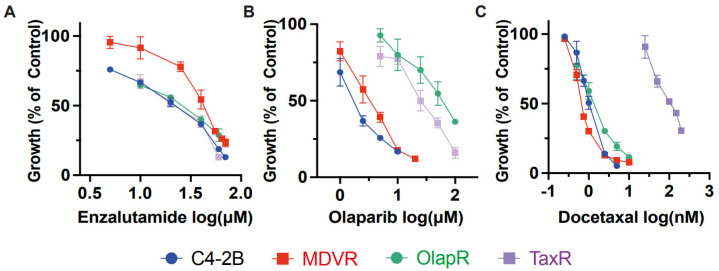
Inter-class resistance across therapeutic classes. (**A**) Cell growth assays were performed to determine the response to enzalutamide, olaparib, and docetaxel in C4-2B parental, MDVR, OlapR, and TaxR cells. A. MDVR cells exhibit resistance whereas OlapR and TaxR display similar responses to enzalutamide. (**B**) Both OlapR as well as TaxR cells display resistance to olaparib treatment. (**C**) TaxR cells solely exhibit resistance to docetaxel, with MDVR and OlapR cells mirroring treatment naïve C4-2B response. Cell growth was quantified as percentage changes to control groups and presented as the mean (±S.D.).

**Figure 4 cancers-15-05273-f004:**
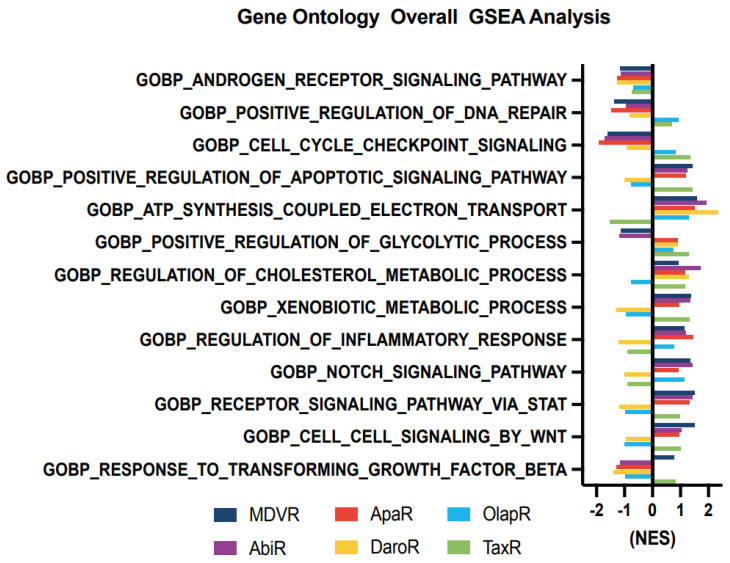
Cell signaling gene enrichment in resistant phenotypes. Gene Set Enrichment Analysis (GSEA) on gene programs relating to Androgen Receptor signaling, DNA repair, cell cycle, metabolism, inflammatory response, and various molecular signaling including NOTCH, STAT and WNT (NES > 1 with *p* < 0.05 considered as significant pathway enrichment). NES, Normalized Enrichment Score.

**Figure 5 cancers-15-05273-f005:**
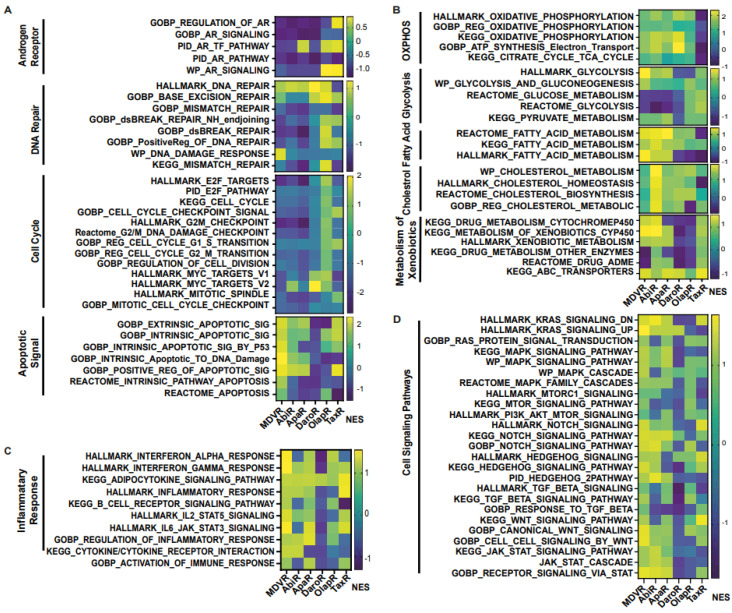
Gene Set Enrichment Analysis (GSEA) indicates variable gene program activation across resistant cell sublines. Cellular processes dysregulated in resistant settings according to the pathway analysis using the Gene Ontology Consortium (GO), the Kyoto Encyclopedia of Genes and Genomes (KEGG), Pathway Interaction Database (PID), WikiPathways (WP), and Reactome Pathways. (**A**) Heatmap of gene set enrichment analysis of androgen receptor, DNA repair, Cell cycle, and apoptotic signaling in resistant cell lines in contrast to parental C4-2B cells. (**B**) Analysis of oxidative phosphorylation, glycolysis, fatty acid and cholesterol regulation, as well as metabolism and clearance of xenobiotics. (**C**) GSEA analysis of genes relating to inflammation and cellular response to immune signaling. (**D**) Enrichment of cellular signaling pathways in resistant cells compared with C4-2B parental cells including KRAS, MAPK, mTOR, etc. (NES > 1 with *p* < 0.05). NES, Normalized Enrichment Score.

**Figure 6 cancers-15-05273-f006:**
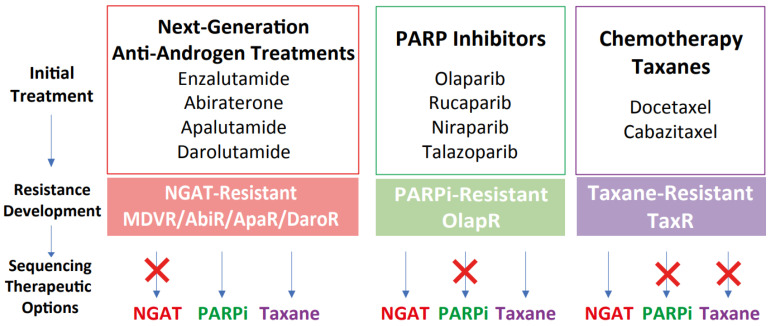
Proposed therapeutic sequencing options based on the models of therapeutic resistance. Schematic illustrating the development of resistance and subsequent therapeutic sequencing options based on the models of acquired resistance. Once resistance developed after NGAT treatments, models suggest subsequent sequencing treatment with either taxanes or PARP inhibitors such as olaparib. For docetaxel resistance, sequencing treatment with olaparib may not be effective in impeding tumor growth, while sequencing to NGAT could be beneficial. For olaparib resistance, sequencing treatment with either NGAT or taxanes may be effective.

## Data Availability

All data generated and collected for analysis in this study are available from the corresponding author upon reasonable request.

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
