# Peer review of "Therapeutic Resistance Models and Treatment Sequencing in Advanced Prostate Cancer"

_cancers, 2023, doi:10.3390/cancers15215273_

Round 1
Reviewer 1 Report
Comments and Suggestions for Authors
The manuscript by Schaaf et al., is focused on the development of strategies to overcome drug resistance in the treatment of prostate cancer. In this study, the authors established multiple castration-resistant prostate cancer (CRPC) models of acquired therapeutic resistance using cell sublines derived from C4-2B, including cell lines resistant to enzalutamide, abiraterone, apalutamide, darolutamide, docetaxel, and Olaparib.
The topic touched upon in the article is relevant. The paper is well-structured, carefully and well written. The scientific content of the manuscript justifies its publication, but some additions and modifications will significantly improve the quality of the article:
- The abstract is very lengthy and goes into detailed accounts that are best suited for the article’s main discussion sections. As such, I suggest the author reduce this section to keep only the most important elements.
- Results: Authors should avoid listing all the values of the results in the text because clearly these can be observed in the figures as well. For example, chapters 3.1 and 3.2 are full of numbers that are already given in the figures.
- Please check the entire text because many typos are present.
Author Response
-

Reviewer 2 Report
Comments and Suggestions for Authors
Castration-Resistant Prostate Cancer (CRPC) is an advanced prostate cancer that keeps growing even though the androgen levels remain low. Common treatments for CRPC constitutes three drug classes: next-generation anti-androgen therapies (NGAT), taxane therapy and PARP inhibitors. Although these treatments have shown efficacy, the continuous emergence of acquired therapeutic resistance remains a major concern. In the manuscript entitled as “Therapeutic resistance models and treatment sequencing in advanced prostate cancer”, Zachary A. Schaaf et al. aim to understand the mechanism underlying drug resistant, and to identify potential therapeutic targets and develop new treatment strategies to overcome drug resistance. To achieve this, the authors collected a series drug-resistant cell sublines against three different classes of drugs, derived from C4-2B cells in their previous study. The authors performed a systematic drug sensitivity assay with different drugs from the three classes among these drug-resistant cell sublines. Their results suggest that intra-class cross-resistance is prevalent within each class of drugs, while the inter-class cross-resistance is rare with the exception that docetaxel resistant cells exhibit reduced sensitivity to olaparib. They proposed a subsequent treatment strategy to counter drug resistance by treating the CRPC cells that are resistant to one of the drug classes with the other two classes of drugs, except for taxane-resistant CRPC, which can only be treated with NGAT."
Moreover, the authors also performed transcriptome sequencing and carried out gene set enrichment analysis (GSEA) in different resistant cell sublines. Their GSEA results uncovered a specific and complex gene expression alteration underlying therapeutic resistance in different drug-resistant cell sublines.
Overall, the study was well-designed, executed very thoroughly, and presented very clearly. The proposed treatment sequencing and the GSEA information will benefit researchers in this field for the development of new drugs and treatment strategies to counter the acquired resistance in CRPC.
Major comments:
1. For ease of reading, please convert all the X-axis coordinates in Figure 2 from logarithmic to linear, as presented in Figure 1.
2. The doses of enzalutamide used in Figure 2A are different among cell lines. Were the results in Figure 2A obtained from the same batch? If not, please consider repeating the experiment with all cell lines in the same batch. Performing drug sensitivity assays in different batches could impact the final results, especially when addressing small differences among different cell lines.
Minor comments:
1. Please swap the positions of Figure 3B and Figure 3C to match the order mentioned in the text.
Author Response
-
